# Effect of Severe Plastic Deformation and Post-Deformation Heat Treatment on the Microstructure and Superelastic Properties of Ti-50.8 at.% Ni Alloy

**DOI:** 10.3390/ma15217822

**Published:** 2022-11-05

**Authors:** Tae-Jin Lee, Woo-Jin Kim

**Affiliations:** Department of Materials Science and Engineering, Hongik University, Mapo-gu, Sangsu-dong 72-1, Seoul 121-791, Korea

**Keywords:** shape memory, superelasticity, aging, severe plastic deformation, grain refinement, superelastic cyclability

## Abstract

Severe plastic deformation via high-ratio differential speed rolling (HRDSR) was applied to the Ni-rich Ti-50.8Ni alloy. Application of HRDSR and a short annealing time of 5 min at 873 K leads to the production of a partially recrystallized microstructure with a small grain size of 5.1 μm. During the aging process for the annealed HRDSR sample at 523 K for 16 h, a high density of Ni_3_Ti_4_ particles was uniformly precipitated over the matrix, resulting in the formation of an *R* phase as the major phase at room temperature. The aged HRDSR sample exhibits excellent superelasticity and superelastic cyclability. This achievement can be attributed to an increase in strength through effective grain refinement and particle strengthening by Ni_3_Ti_4_ and a decrease in the critical stress for stress-induced martensite (B19′) due to the presence of the *R*-phase instead of B2 as a major phase at room temperature. The currently proposed method for using HRDSR and post-deformation heat treatment allows for the production of Ni-rich NiTi alloys with excellent superelasticity in sheet form.

## 1. Introduction

Ni-rich NiTi alloys containing more than 50.6 at.% nickel exhibit superior superelasticity [1,2]. Superelasticity occurs at temperatures above the austenite finish temperature (*A_f_*) upon loading, and a stress hysteresis forms in the tensile or compressive stress–strain curve due to phase transformation during loading and unloading [1,2]. Superelasticity of NiTi alloys has been extensively studied for applications in biomedical and engineering fields [1,3,4,5]. However, control of the superelastic properties of Ni-rich NiTi alloys is difficult because the characteristics of the phase transformation sensitively vary with small changes in microstructure and composition [2]. The shape memory functions of Ni-rich NiTi alloys are known to be greatly affected by aging due to the formation of Ni_4_Ti_3_ precipitates [6,7,8]. After aging, the martensite start temperature (*M_s_*) and austenite finish temperature (*A_f_*) tend to increase with increasing aging time due to Ni depletion of the matrix by precipitation of the Ni_4_Ti_3_ phase [9]. Thus, aging is an effective way of controlling the superelastic characteristics of Ni-rich NiTi alloys. Grain size also greatly affects the transformation stress, transformation strain and hysteresis loop area of Ni-rich NiTi alloys [10,11,12,13]. From a mechanical viewpoint, hardening by Ni_4_Ti_3_ precipitates and grain-size reduction in Ni-rich NiTi alloys can increase the critical stress for the occurrence of slip, which is beneficial for achieving good superelasticity. This is because if the critical stress for slip is low, there is a high chance of slip by dislocations during loading.

Recently, interest in superelasticity originating from multistep phase transformation has increased [14,15,16]. During the multistep transformation process, an intermediate phase, the *R*-phase with a rhombohedral crystal structure that grows mainly on the {111} plane of B2 austenite, is formed, showing a two-step change for the phase structure: B2 → *R* → B19′ [15,16] or a three-step change for the phase structure: B2 → *R1*, B2 → *R2*, *R1*/*R2* → B19′ [7] upon cooling. The *R*-phase transformation can be introduced depending on the thermomechanical treatment conditions [7,14,15,16]. During cooling, *R* is formed before the formation of B19′ because B19′ is thermodynamically preferred, but the *R* phase is kinetically advantageous, having low activation energy [17]. As the B2 to *R* phase transformation accompanies a significantly smaller transformation strain (less than 1% [16,17,18]) compared to the B2 to B19′ phase transformation (approximately 8–10% [17,19,20]), this characteristic is useful for small amplitude but higher frequency actuator and damping applications [21].

Heavy plastic deformation or severe plastic deformation (SPD) has been demonstrated to be effective in the grain refinement of NiTi alloys [22,23,24,25,26,27,28,29,30]. To date, many studies have been conducted to investigate the effect of grain size on superelastic properties [28,29,30]. Malard et al. [28] and Delville et al. [29] showed that a Ni-rich NiTi alloy, which has been cold worked and then annealed by a pulsed electric current, is highly resistant to dislocation slip at grain sizes < 100 nm, while that with fully recrystallized microstructures and grain sizes exceeding 200 nm is prone to dislocation slip. The authors showed that grain refinement to the grain size <100 nm renders the alloy have superior superelasticity. Tong et al. [30] applied equal channel angular pressing (ECAP) to a Ni-rich NiTi alloy with subsequent annealing at 573–873 K. The sample with a grain size of 0.3 μm exhibited the best superelasticity and cycling stability.

Superelasticity of NiTi alloys is not often determined by a single microstructural factor but by a combined effect involving the grain size, Ni_4_Ti_3_ precipitate, texture and dislocation density. The combined effect of grain-size reduction by SPD and aging on the superelasticity of Ni-rich NiTi alloys has been relatively rarely studied. In this work, a Ni-rich NiTi alloy was processed by high-ratio differential speed rolling (HRDSR), which is a severe plastic deformation method applicable for materials in sheet form [22,31], and aging was applied to the deformed samples. The effects of grain size, dislocation density, texture and Ni_4_Ti_3_ precipitates on phase transformation and superelastic behavior were examined.

## 2. Experimental Procedures

Ti-50.8% at. Ni alloy plates with 3 mm × 2.6 mm × 100 mm were purchased from SMA Co., Ltd (Seoul, Republic of Korea). The purchased plate was heat-treated at 1023 K for 15 min to relieve any residual stress. This sample will be referred to as the as-received (AR) sample. For applying severe plastic deformation at cryogenic temperatures, which is known to be more effective in microstructural refinement compared to SPD at room temperature [22,32], the AR sample was immersed into liquid nitrogen for 10 min, removed from the liquid nitrogen bath and then immediately subjected to differential speed rolling with a speed ratio of 2:1 between the upper and lower rolls. After two passes, the thickness was reduced to a value of 1.55 mm, corresponding to a thickness reduction of 40%. This sample will be hereafter referred to as the HRDSR sample. The AR and HRDSR samples were annealed for 5 min and 120 min at a temperature of 873 K, respectively, in an argon atmosphere. Some of the annealed AR and HRDSR samples were aged for 16 h at 523 K in an argon atmosphere.

To evaluate the superelastic properties of the AR, HRDSR, annealed AR, annealed HRDSR, aged AR and aged HRDSR samples, a cyclic test through tensile loading–unloading was performed. A 6% strain was applied at a test temperature of 298 K.

The microstructure of the samples was observed using field emission scanning electron microscopy (BSE of FE-SEM (SU-5000, Hitachi, Tokyo, Japan), field emission transmission electron microscopy (FE-TEM (JEM 2001 F, 200 keV, JEOL, Tokyo, Japan) and electron backscattering diffraction (EBSD) analysis (Velocity Super, EDAX, Mahwah, United States). For the FE-TEM observation, samples were jet polished with a solution composed of 60% methyl alcohol (CH_3_OH), 30% glycerin (C_3_H_8_O_3_) and 10% nitric acid (HNO_3_) and then ion milled. For the EBSD observation, the sample with a transverse cross section along the RD was mechanically ground using SiC paper and polished using 3 μm and 1 μm diamond suspensions and OP-S suspensions in order. EBSD data were analyzed with a step size of 0.2 μm using TSL-OIM analysis software, excluding data with a confidence index value of 0.1 or less. The average grain size was determined with a grain tolerance angle of 5°, and the fraction of recrystallized grains was determined using the grain orientation spread (GOS) method. The GOS of a recrystallized grain was assumed to be less than 2°.

A differential scanning calorimeter, DSC (DSC 200 F3 Maia, NETZSCH, Selb, Germany), was used in the temperature range of 123 K to 373 K at a heating and cooling rate of 5 K/min to determine the phase transformation temperature.

High-resolution X-ray diffraction (HR-XRD, SmartLab, Rigaku, Tokyo, Japan) with a CuKa (λ = 1.5412 Å) target was used for phase identification of the deformed and heat-treated samples at scan angles ranging from 35 to 80 degrees.

## 3. Results

### 3.1. Initial Material

**Figure 1a**–**f** show the EBSD inverse pole figure (IPF), grain boundary (GB) and kernel average misorientation (KAM) maps for the AR and HRDSR samples. The grain size of the AR sample is 10.9 μm. The microstructure of the AR sample is composed of equiaxed grains, some of which contain dislocation substructure. The HRDSR sample shows a heavily deformed microstructure. The fraction of low-angle grain boundaries is as high as 0.86. The <110>//ND_B2_ texture in the austenite phase of the AR sample is retained after deformation by HRDSR. 

**Figure 2a** shows the DSC results for the AR and HRDSR samples. The phase transformation temperatures determined from the DSC curves are summarized in **Table 1**. The AR sample exhibits a two-stage phase transformation (B2 austenite → *R*-martensite → B19′ martensite) upon cooling. The *R*-phase peak is relatively small and broad compared to the B19′ peak. Upon heating, the peak associated with the *R*-phase does not appear, and only a single phase transformation from B19′ to B2 is observed. This type of asymmetric *R*-phase transformation is known to occur when B19′ is energetically preferred over *R* at all temperatures, but *R*-phase formation has a lower kinetic barrier upon cooling [33]. The HRDSR sample does not show any phase transformation during heating and cooling, indicating that the introduction of a high dislocation density by SPD suppresses the martensitic transformation upon cooling.

**Figure 2b** shows the XRD curves for the AR and HRDSR samples, respectively. For both samples, only the B2 austenite phase is identified, which agrees with the prediction from the DSC result. Compared to the AR sample, the HRDSR sample has considerably broader peaks, indicating that the dislocation density and the ratio of amorphous and nanosized grains are greatly increased after SPD by HRDSR.

### 3.2. Materials Processed by HRDSR

#### 3.2.1. Microstructures

**Figure 3a**–**h** shows the EBSD inverse pole figure maps for the annealed and aged AR and HRDSR samples. The grain size and fraction of recrystallized grains determined based on the EBSD data are plotted in **Figure 4a,b**, respectively. The grain size is increased with annealing time. For the AR sample, the grain size is increased from 10.9 to 13.9 μm after annealing for 120 min. For the HRDSR sample, the grain sizes after annealing for 5 min and 120 min are 5.1 μm and 8.7 μm, respectively. During the subsequent aging process, noticeable grain growth occurs in the AR and HRDSR samples annealed for 5 min, while limited grain growth occurs in the AR and HRDSR samples annealed for 120 min. The fraction of recrystallized grains in the AR sample increases from 0.33 to 0.43 and 0.78 after annealing for 5 and 120 min, respectively. For the HRDSR sample, the fraction of recrystallized grains dramatically increases after 5 min of annealing (from 0.03 to 0.68) and is further increased to 0.78 after annealing for 120 min. Inverse pole figures, given as insets, show that the <110>//ND_B2_ texture component in the AR sample is retained after annealing and aging. For the HRDSR sample, after annealing, a new texture component (<111>//ND)_B2_ develops, supporting the occurrence of recrystallization during the heat treatment, and this texture component retains after aging. For the aged AR and HRDSR samples, the fraction of recrystallized grains decreases (rather than increases) after aging (except for the AR sample annealed for 5 min), and this decrease is most pronounced in the HRDSR sample annealed for 5 min. This unexpected result is most likely due to the precipitation of Ni_4_Ti_3_ particles that creates coherency strain fields surrounding them and induces the formation of *R*-phase around the particles within grain interiors, leading to an increase in GOS to a value of over 2° in the recrystallized grains obtained during the annealing process. This will be discussed later.

#### 3.2.2. Phase Transformation Temperatures

**Figure 5a**–**c** shows the DSC results for the annealed and aged AR and HRDSR samples. The phase transformation temperatures determined from the DSC curve are summarized in **Table 1**. The HRDSR sample, which does not show the austenite–martensite transformation peak upon cooling (**Figure 2a**), shows a small and broad austenite–martensite transformation peak after annealing for 5 min (**Figure 5a**). The B2→B19′ transformation becomes more obvious after prolonged annealing for 120 min. This observation indicates that the increased degree of recrystallization promotes the B2–B19′ transformation because the density of dislocations is further decreased with increasing amount of recrystallization. After aging, two-stage transformations occur in both the AR and HRDSR samples. Two-stage transformation occurs upon cooling as well as heating in both alloys. This type of symmetric *R*-phase transformation is known to occur when there is a temperature window in which *R* is thermodynamically favored over both B2 and B19′ [33]. The peaks for the B2 → *R* and *R* → B2 transformations in all the aged samples are located at approximately 300 K and 315 K upon cooling and heating, respectively. It is noted from the DSC curves that the transformation temperatures for B2 → *R* and *R* → B2 are less sensitive to the microstructure compared to the transformation temperatures for *R*→B19′ and B19′ → *R*. **Figure 5c** shows the magnified DSC curves for the aged AR and HRDSR samples during cooling in the temperature range between 220 and 300 K. Unlike the aged AR samples, small peaks appear in the aged HRDSR samples. This result suggests the possibility of the occurrence of B2 → *R*2 transformation in the aged HRDSR samples. It was claimed that the phase transformation of B2 → *R*1 occurs due to the generation of Ni_4_Ti_3_ near high-energy grain boundaries, and B2 → *R*2 phase transformation occurs when the dislocation networks existing inside the grains act as nucleation sites for Ni_4_Ti_3_ precipitation [6,7].

#### 3.2.3. Precipitates

**Figure 6a,b** shows the XRD curves for the annealed and aged AR and HRDSR samples. Only the B2 austenite phase is observed in both the annealed AR and HRDSR samples. After aging treatment, *R*-phase peaks can be observed, which is clearly evidenced by a splitting of the (1 1 0) austenitic peak [7] in both the AR and HRDSRed samples, indicating that the microstructures of the aged AR and HRDSR samples are composed of a mixture of B2 and *R* phases at room temperature, in agreement with the result expected from the DSC curves at 298 K.

**Figure 7a**–**h** shows the KAM maps for the annealed and aged AR and HRDSR samples, and **Figure 8** shows a plot of their average KAM values. The KAM value indicates the average misorientation value between the measurement point and its surrounding points, which indicates the local strain [34]. Therefore, the density of geometrically necessary dislocations, which are necessary for preserving lattice continuity, increases with the KAM value [35]. The KAM value decreases after annealing for 5 min in both the AR and HRDSR samples, but a more significant decrease occurs in the HRDSR sample, which is due to the occurrence of recrystallization in the HRDSR sample. A large increase in KAM after aging occurs in the HRDSR sample annealed for 5 min; however, the KAM value does not change much after aging of the AR sample annealed for 5 min. The large increase in the KAM value over the matrix of the HRDSR sample after aging can be attributed to precipitation of a large amount of Ni_4_Ti_3_ particles during aging, which occurs because the large grain boundary areas and dislocation substructure within interiors of small partially recrystallized grains provide the preferred sites for nucleation of Ni_4_Ti_3_ particles and fast atomic diffusion paths. The formation of the Ni_4_Ti_3_ precipitates promotes the formation of the *R* phase by impeding the B2→B19′ transformation, but as Ni_4_Ti_3_ creates high-strain energy, the nucleation and growth of *R* preferentially occur near the grain boundary and dislocation substructure where the nucleation barrier for the *R* phase is low [36]. The fraction of low-angle grain boundaries of the HRDSR sample (annealed for 5 min) has increased from 0.23 to 0.75 after aging, and this is most likely due to the creation of many interfaces between the *R* and B2 phases. For the HRDSR sample annealed for 120 min, the increase in the average KAM value after aging is relatively small, and the large KAM values are confined to near grain boundaries. This results because the grain boundary area and the density of dislocation substructure largely decreased during the long-time annealing of 120 min. 

**Figure 9a,b** shows TEM micrographs for the aged HRDSR samples (after annealing for 5 min). It is evident that Ni_4_Ti_3_ with a typical lenticular shape is nucleated within grain interiors as well as near grain boundaries. The *R*-phase is often observed near Ni_4_Ti_3_ phases, supporting that Ni_4_Ti_3_ phase promotes the occurrence of the *R* phase. **Figure 9c** shows a TEM micrograph of the aged HRDSR sample (after annealing for 120 min). It is noted that the size of the Ni_4_Ti_3_ precipitate is larger than that observed for the aged HRDSR sample (after annealing for 5 min). **Figure 9d** shows an SEM micrograph of the aged HRDSR sample (after annealing for 120 min), where most of the Ni_4_Ti_3_ particles are observed to nucleate and grow near grain boundaries. At the same SEM magnification, it is hard to find Ni_4_Ti_3_ particles in the matrix of the aged HRDRS sample (after annealing for 5 min) (not shown here), indicating that Ni_4_Ti_3_ precipitate particles in the aged HRDRS sample (after annealing for 5 min) are much smaller than those in the aged HRDRS sample (after annealing for 120 min).

#### 3.2.4. Texture

The superelastic strain of NiTi alloys depends on the crystal orientation. Miyazaki et al. [37] showed that [233]_B2_, [111]_B2_ and [011]_B2_ have high superelastic strains of 10.7%, 9.8% and 8.4% in a single crystal [3,37,38]. **Figure 10** shows the [233]_B2_//RD, [111]_B2_//RD, and [011]_B2_//RD texture components mapped on the EBSD-generated microstructures of the AR and HRDSR samples and **Figure 11** shows the total fractions of grains with the three texture components in the AR and HRDSR samples. The calculated fractions for the annealed and aged AR and HRDSR samples range between 0.4 and 0.6, but the HRDSR samples have lower fractions than the AR samples. This result implies that from a texture viewpoint, the HRDSR samples do not have an advantage in achieving better superelasticity compared to the AR samples.

#### 3.2.5. Mechanical Properties

**Figure 12a,b** shows the Vickers hardness measurement results obtained for the AR and HRDSR samples after annealing and aging. For the AR sample, after annealing for 5 min, the hardness is slightly decreased to 259.1 Hv due to the annealing effect. However, upon subsequent aging, the hardness is increased to 280–288 Hv. The HRDSR sample shows a Hv of 354, which is significantly higher than that for the AR sample (266.8 Hv). After annealing for 5 min, the hardness is largely decreased to 251 Hv due to the occurrence of recrystallization. Upon subsequent aging, however, the hardness is increased to 327.2–342.9 Hv, which is much higher than that obtained for the AR sample aged under the same conditions. The hardness of the HRDSR sample annealed for 120 min also increases after aging, but the obtained hardness is notably lower than that of the aged HRDSR sample (after annealing for 5 min).

**Figure 13a**–**f** shows the cyclic tensile test results for the annealed AR and HRDSR samples up to 6%. The measured residual strains are plotted as a function of the number of cycles in **Figure 14**. For both the annealed AR and HRDSR materials, B19′ is expected to be directly induced from B2 during loading. For the AR sample, annealing treatment for 5 min reduces the residual strain from 3.25% to 1.4% in the first cycle, indicating a positive effect of annealing on superelasticity. As the number of cycles increases, the residual strain increases and then tends to become saturated after many cycles. For the HRDSR sample, fracture occurs before reaching a strain of 6% due to a significant deterioration of ductility after heavy plastic deformation, which is typical in many SPD-processed metals [2]. After annealing for 5 min and 120 min, the residual strain after the first cycle is approximately 2% and fracture occurs after 4~6 cycles. There is a notable difference between the annealed AR and HRDSR samples in terms of the critical stress for stress-induced martensitic transformation: the critical stresses of the annealed AR samples are lower than those for the annealed HRDSR samples (357.0–375.1 MPa vs. 433.0–437.0 MPa). This difference can be attributed to the grain-size effect on critical stresses for stress-induced martensitic transformation because, as the grain size becomes smaller, the barrier for martensitic transformation is expected to become higher [11].

**Figure 15a**–**d** shows the cyclic tensile test results for the aged AR and HRDSR samples up to 6%. The measured residual strains are plotted as a function of the number of cycles in **Figure 14**. A significant improvement in superelasticity is observed in both materials compared to the cases when only annealing is applied. Unlike the annealed materials with the B2 phase, the aged materials contain a mixture of B2 and *R* phases, and the amount of *R* is expected to be largest in the aged HRDSR sample (after annealing for 5 min). The slope for elastic deformation of the aged HRDSR sample (after annealing for 5 min) is apparently lower than that of the aged HRDSR samples (after annealing for 120 min) as well as the aged AR samples. This is because the elastic modulus of the *R* phase is lower than that of B2 (20 [16] vs. 60–70 GPa [39]) such that the elastic modulus of the aged HRDSR sample (after annealing for 5 min) with the *R*-phase as a major phase is relatively low. For the aged HRDSR sample (after annealing for 5 min), where B19′ is expected to be stress induced from *R* rather than B2 during loading and that B19′ reverts to *R* during unloading, the residual strain remains virtually zero after many cycles. This result indicates that the aged HRDSR sample (after annealing for 5 min) exhibits excellent superelasticity and cyclic stability. For the aged HRDSR sample (after annealing for 120 min), where B19′ is expected to be stress induced from B2 during loading, the residual strain is larger than that for the aged HRDSR sample (after annealing for 5 min) from the first cycle and with repeated cycling, the residual strain continues to increase and then saturates beyond five cycles. Compared to the aged HRDSR samples, the aged AR samples exhibit poorer superelasticity and cyclability. It is worthwhile to note that the aged HRDSR sample (after annealing for 5 min) exhibits incremental variation in stress at plateau during repeated superelastic loading and unloading, while the other samples show a flat stress plateau. Wang et al. [40] also observed the incremental stress variation during superelastic loading in the swaged NiTi and attributed this phenomenon to the heterogeneous microstructure of the swaged sample (mixed with high- and low-angle grain boundaries) where martensitic transformation occurs first at high-angle grain boundaries and then later at low-angle grain boundaries. The microstructure of the aged HRDSR sample (after annealing for 5 min) also consists of many high-angle grain boundaries and dislocation substructures in grain interiors.

From **Figure 15a**–**d**, it is also recognized that the aged HRDSR sample (after annealing for 5 min) shows the lowest critical stress for martensitic transformation among the four aged samples, even though it has the smallest grain size. This is most likely because the barrier for transformation from *R* to B19′ is smaller than that for transformation from B2 to B19′ [17]. The aged HRDSRed sample (after annealing for 5 min) also exhibits the smallest hysteresis (low dissipation energy).

## 4. Discussion

As the aged HRDSR samples do not have a favorable texture compared to the aged AR samples, the superior superelasticity of the former should originate from their microstructures. Tong et al. [41] proposed that the difference between the yield strength of B2 and the critical stress for the phase martensitic transformation (Δσ) is related to the recovery strain, which is equal to the applied strain minus the residual strain. Here, the yield stresses for the annealed and aged AR and HRDSR samples were estimated based on their Vickers hardness data using the relation of $\sigma_{y}=3.03H_{v}$ [42], where $\sigma_{y}$ is the yield stress (MPa) and $H_{v}$ is the Vickers hardness (kg/mm^2^). **Figure 16**a shows the Δσ calculated using the $\sigma_{y}$ values calculated from the $H_{v}$ data and the critical stresses for the martensitic transformation measured from the tensile tests in **Figure 13 and Figure 15**. The aged HRDSRed sample (after annealing for 5 min) exhibits the largest Δσ. This is because the critical stress for phase transformation from *R* to B19′ is lower than that from B2 to B19′ and the yield strength is high due to effective grain refinement and particle strengthening. 

**Figure 16b** shows the relationship between Δσ and residual strain. There is a trend that as Δσ increases, the residual strain decreases. This result suggests that the HRDSR technique can greatly enhance the superelasticity of Ni-rich NiTi alloys by increasing the strength (against slip) through effective grain refinement and aging and by decreasing the critical stress for stress-induced martensite (by having the *R* phase as a major phase prior to loading). Reduction of the dislocation density by annealing is important because the presence of a high dislocation density disturbs the phase transformation from B2⟷B19′ or *R*⟷B19′ and reduces the cyclic number. However, when the dislocation density or dislocation substructure is reduced too much through long-term annealing, the nucleation sites for Ni_3_Ti_4_ precipitates can be greatly reduced, leading to a decrease in the yield strength and an increase in the critical stress for stress-induced martensite by decreasing the volume fraction of the *R*-phase at room temperature under unstressed condition. 

## 5. Conclusions

The combined effect of grain-size reduction by SPD and aging on the superelasticity of Ni-rich NiTi alloy was studied, and the following results were obtained:^1.^ Severe plastic deformation by HRDSR and subsequent short-term annealing for 5 min at 873 K produces a partically recrystallized microstructure with a small grain size of 5.1 μm.^2.^ During the aging of the annealed HRDSR sample at 523 K for 16 h, a high density of Ni_3_Ti_4_ particles is densely and uniformly precipitated over the matrix, resulting in the formation of an *R* phase as the major phase at room temperature. For a long annealing time before aging, the dislocation substructure within the grain interiors is diminished, and the grain boundary area decreases, such that the precipitation of Ni_3_Ti_4_ during aging is small, and their distribution is inhomogeneous.^3.^ The difference between the yield strength and critical stress for the stress-induced martensitic transformation (Δσ) is found to be closely related to the superelastic strain. As Δσ increases, the superelastic strain increases.^4.^ Superelasticity and cyclability of a Ni-rich NiTi alloy can be enhanced by increasing the strength through effective grain refinement via SPD plus annealing and aging for precipitation of Ni_3_Ti_4_ and by decreasing the critical stress for stress-induced martensite through incorporation of the *R*-phase as a major phase at room temperature.

## Figures and Tables

**Figure 1 materials-15-07822-f001:**
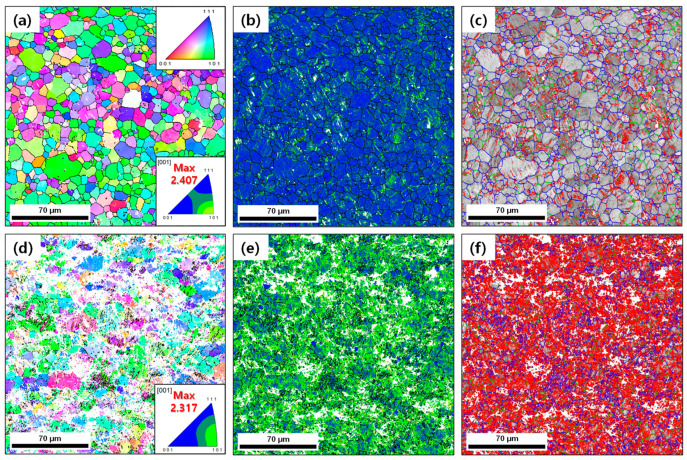
The inverse pole figure, KAM and GB maps for the (**a**–**c**) as-received (AR) and (**d**–**f**) HRDSR samples. In the GB map, low-angle boundaries (2–5°) are in red, intermediate angle boundaries (5–15°) are in green and high-angle boundaries (>15°) are in blue.

**Figure 2 materials-15-07822-f002:**
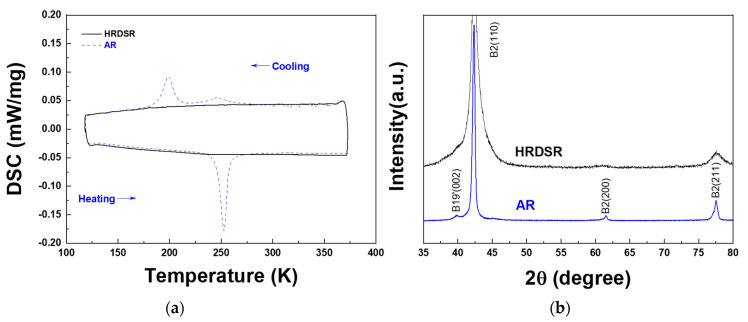
(**a**) The DSC curve for the AR and HRDSR samples. (**b**) The XRD curves for the AR and HRDSR samples. Identification of phases was made based on the data from JCPDF cards (01−076 −3614, 01−076−7519 and 01−076−4263).

**Figure 3 materials-15-07822-f003:**
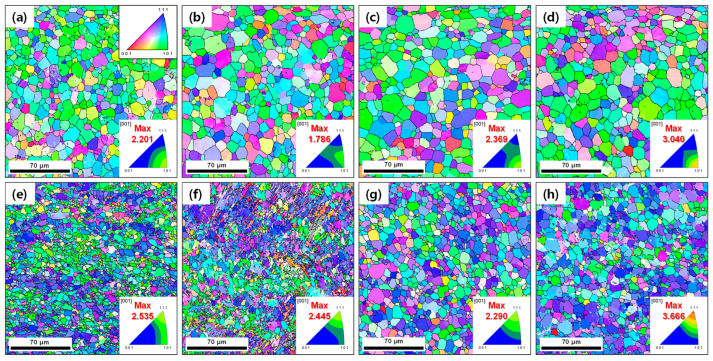
The IPF maps for the AR samples (**a**) annealed for 5 min at 873 K, (**b**) annealed for 5 min at 873 K and then aged at 523 K for 16 h, (**c**) annealed for 120 min at 873 K, (**d**) annealed for 120 min at 873 K and then aged at 523 K for 16 h. The IPF maps for the HRDSR samples (**e**) annealed for 5 min at 873 K, (**f**) annealed for 5 min at 873 K and then aged at 523 K for 16 h, (**g**) annealed for 120 min at 873 K, (**h**) annealed for 120 min at 873 K and then aged at 523 K for 16 h.

**Figure 4 materials-15-07822-f004:**
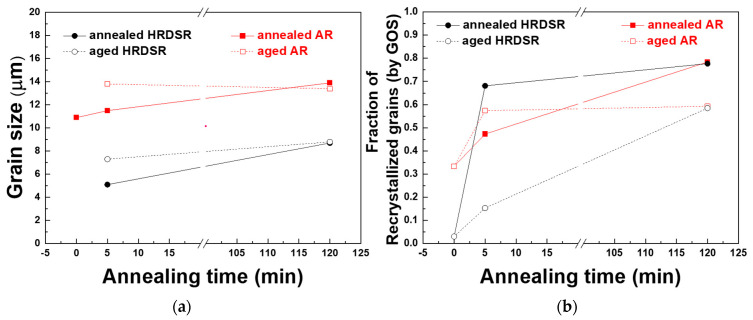
(**a**) The grain size of the AR and HRDSR samples before and after heat treatment. (**b**) The fraction of recrystallized grains in the AR and HRDSR samples before and after heat treatment determined using GOS.

**Figure 5 materials-15-07822-f005:**
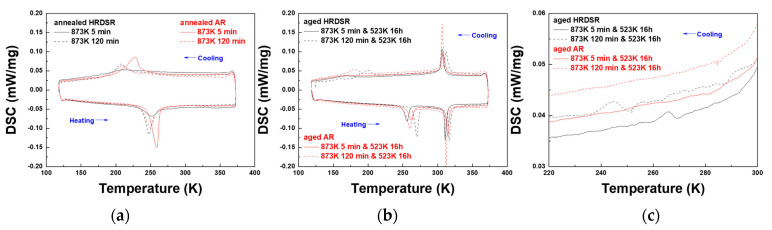
The DSC curves for (**a**) the annealed and (**b**) the aged AR and HRDSR samples. (**c**) The DSC curves for the aged AR and HRDSR samples magnified in the temperature range between 220 and 300 K upon cooling.

**Figure 6 materials-15-07822-f006:**
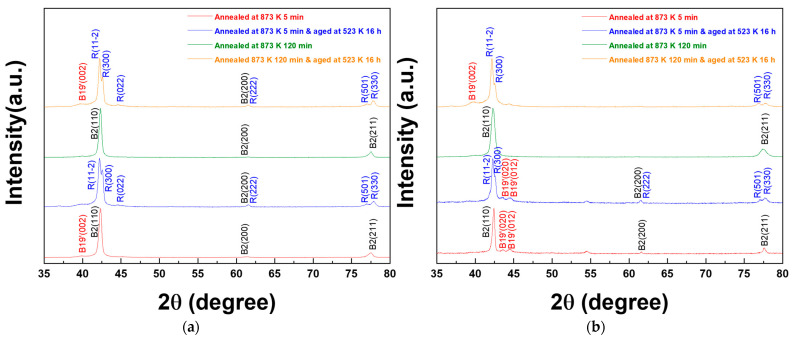
The XRD curves for the (**a**) AR and (**b**) HRDSR samples after annealing or annealing plus aging. Identification of phases was made based on the data from JCPDF cards (01−076−3614, 01−076−7519 and 01−076−4263).

**Figure 7 materials-15-07822-f007:**
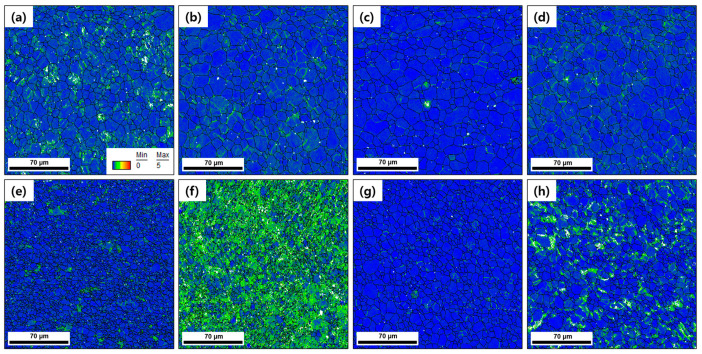
The KAM maps for the AR samples (**a**) annealed for 5 min at 873 K, (**b**) annealed for 5 min at 873 K and then aged at 523 K for 16 h, (**c**) annealed for 120 min at 873 K, (**d**) annealed for 120 min at 873 K and then aged at 523 K for 16 h. The IPF maps for the HRDSR samples (**e**) annealed for 5 min at 873 K, (**f**) annealed for 5 min at 873 K and then aged at 523 K for 16 h, (**g**) annealed for 120 min at 873 K, (**h**) annealed for 120 min at 873 K and then aged at 523 K for 16 h.

**Figure 8 materials-15-07822-f008:**
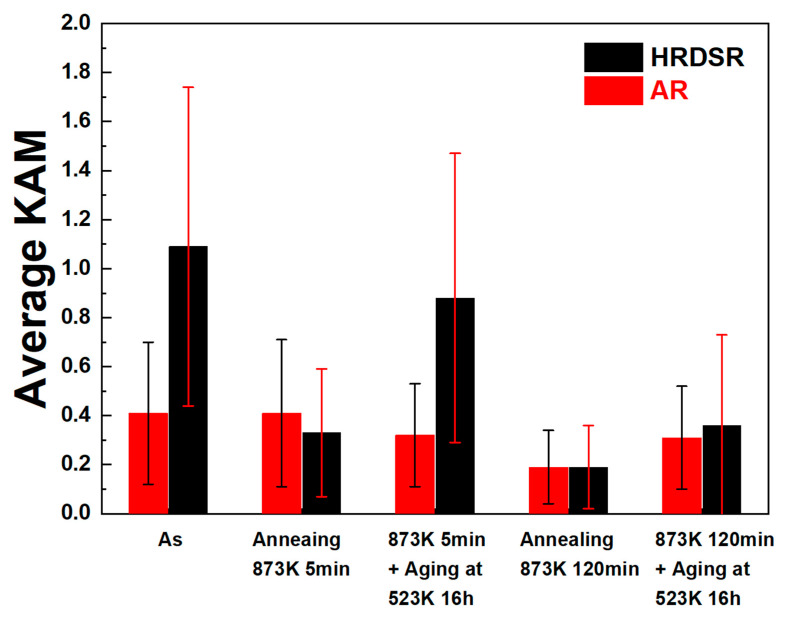
The average KAM values of the AR and HRDSR samples after annealing or annealing plus aging.

**Figure 9 materials-15-07822-f009:**
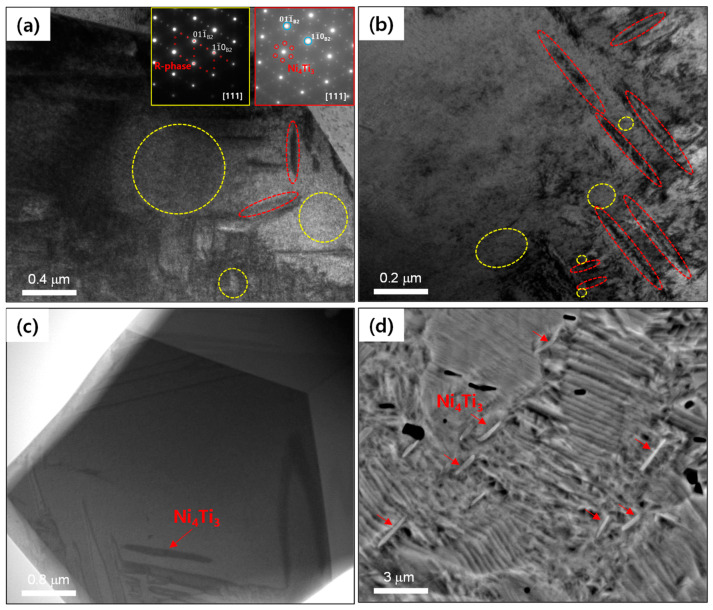
The TEM micrographs for the aged HRDSR sample (after annealing for 5 min): (**a**) near grain boundaries and (**b**) grain interior. The (**c**) TEM and (**d**) SEM micrographs for the aged HRDSR sample (after annealing for 120 min). The regions marked by yellow and red circles represent the regions where B2+*R* phases and B2+Ni_4_Ti_3_ phases are identified to exist, respectively.

**Figure 10 materials-15-07822-f010:**
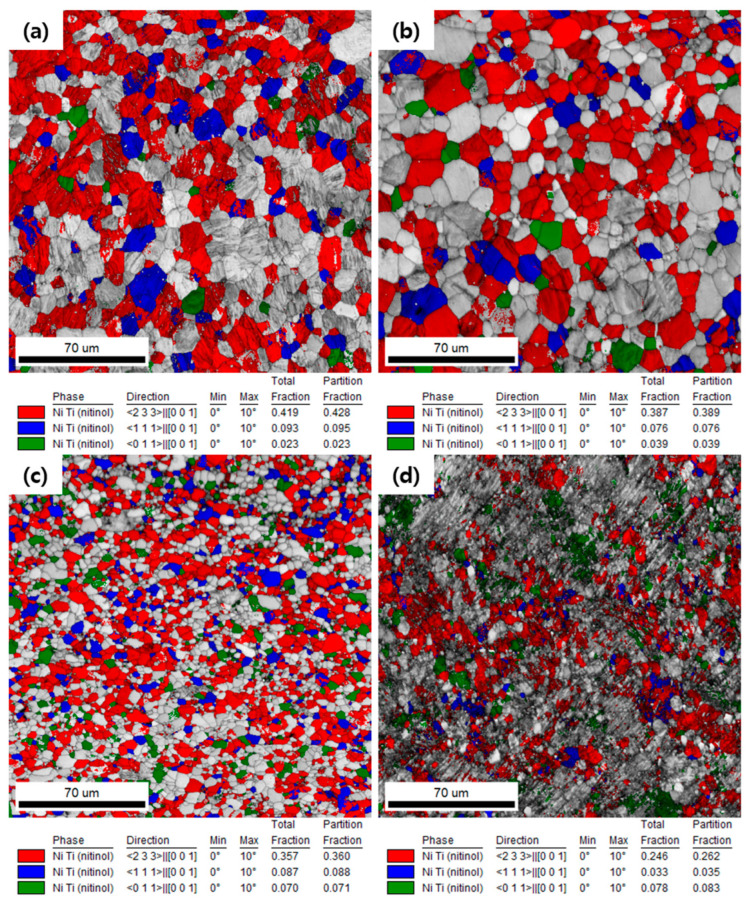
The [233]_B2_//RD, [111]_B2_//RD and [011]_B2_//RD texture components mapped on the EBSD-generated microstructures of the AR samples (**a**) annealed for 5 min at 873 K and (**b**) annealed for 5 min at 873 K and then aged at 523 K for 16 h, and the HRDSR samples (**c**) annealed for 5 min at 873 K and (**d**) annealed for 5 min at 873 K and then aged at 523 K for 16 h.

**Figure 11 materials-15-07822-f011:**
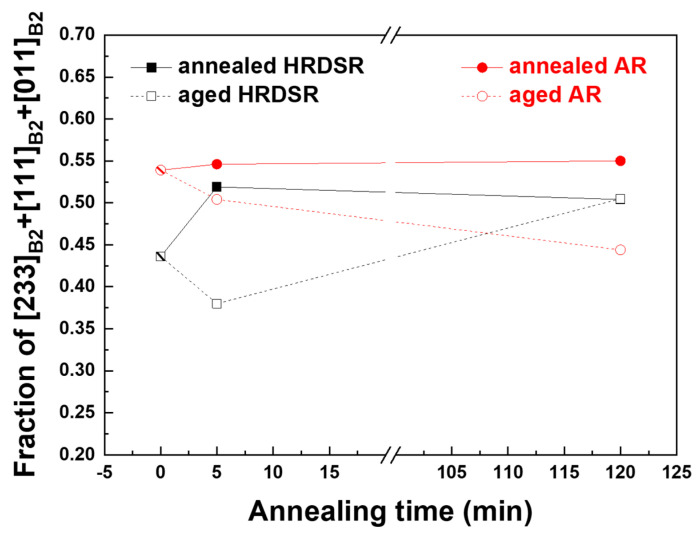
The total fraction of grains with the texture components of [233]_B2_//RD, [111]_B2_//RD and [011]_B2_//RD texture components for the AR and HRDSR samples after annealing or annealing plus aging.

**Figure 12 materials-15-07822-f012:**
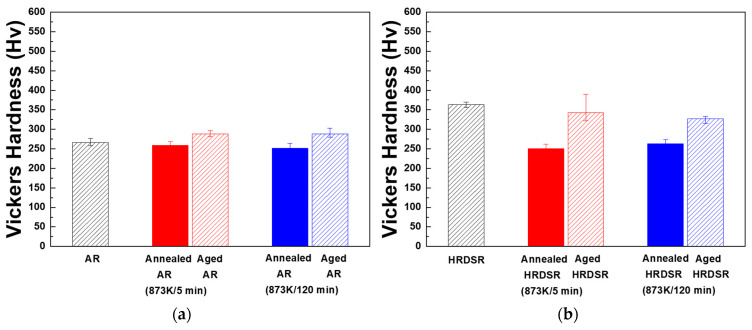
The Vickers hardness of the (**a**) AR and (**b**) HRDSR samples after annealing or annealing plus aging.

**Figure 13 materials-15-07822-f013:**
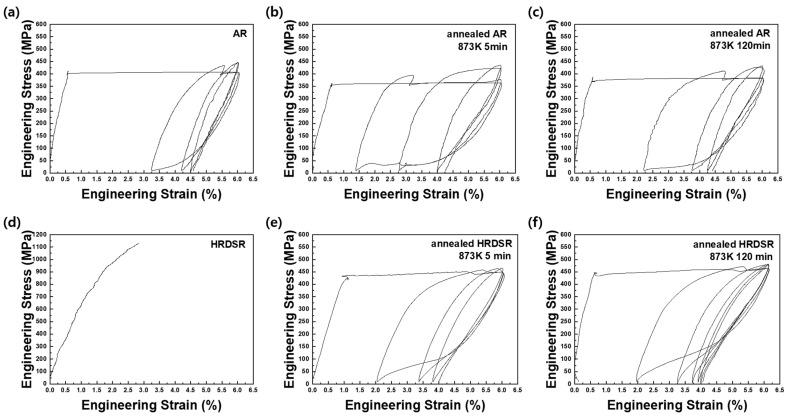
Superelastic cyclic tests up to the strain of 6% for (**a**) the AR sample, (**b**) the AR sample annealed at 873 K for 5 min, (**c**) the AR sample annealed at 873 K for 120 min, (**d**) the HRDSR sample, (**e**) the HRDSR sample annealed at 873 K for 5 min and (**f**) the HRDSR sample annealed at 873 K for 120 min.

**Figure 14 materials-15-07822-f014:**
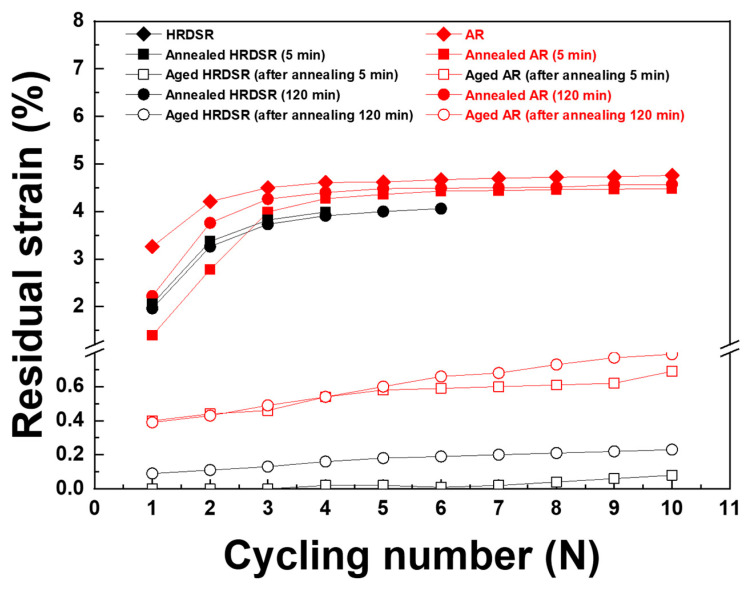
Residual strain as a function of cyclic number for the AR and HRDSR samples after annealing or annealing plus aging.

**Figure 15 materials-15-07822-f015:**
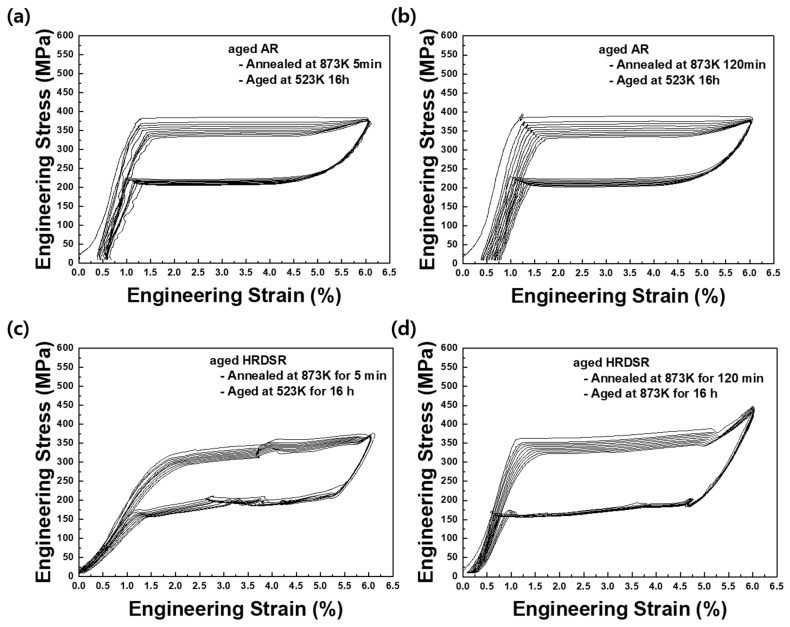
Superelastic cyclic tests up to the strain of 6% for (**a**) the aged AR sample (after annealing at 873 K for 5 min), (**b**) the aged AR sample (after annealing at 873 K for 120 min), (**c**) the aged HRDSR sample (after annealing at 873 K for 5 min) and (**d**) the aged HRDSR sample (after annealing at 873 K for 120 min).

**Figure 16 materials-15-07822-f016:**
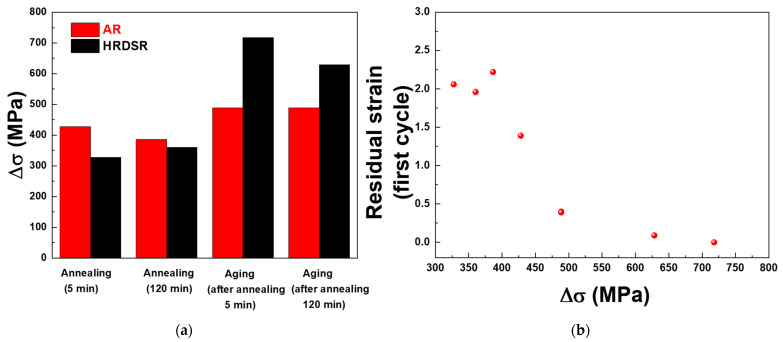
(**a**) The difference between the yield strength and the critical stress for phase martensitic transformation (Δσ) and (**b**) the relationship between Δσ and residual strain.

**Table 1 materials-15-07822-t001:** DSC test results for the AR and HRDSR samples before and after various heat treatments.

Samples	Transformation Temperature (K)							
	Cooling				Heating			
	*R_s_*	*R_f_*	*M_s_*	*M_f_*	*R_s_*	*R_f_*	*A_s_*	*A_f_*
**As-purchased**			246.7	227.3			262.1	277.7
**AR**	268.9	229.3	207.7	187.3	-	-	245.9	257.5
**annealed at 873 K for**	□				□			
**5 min**	-	-	236.1	204.7	-	-	245.4	264.4
**120 min**	-	-	215.6	208.2	-	-	236.2	258.4
**5 min + aged at 523 K**	313.5	303.5	189.2	136.2	253.8	267.4	311.1	318.3
**120 min + aged at 523 K**	309.4	304.2	189.3	159.3	253.3	265.1	310.7	313.7
**HRDSR**	-	-	-	-	-	-	-	-
**annealed at 873 K for**	□				□			
**5 min**	-	-	274.7	181	-	-	218.5	267.6
**120 min**	-	-	217.7	195.8	-	-	238.8	256
**5 min + aged at 523 K**	310.2	303.5	189.9	137.4	249.6	260.6	308.9	313.9
**120 min + aged at 523 K**	322.3	308.4	210.6	187.8	264.9	273.8	314.5	319.6

## Data Availability

The raw/processed data required to reproduce these findings cannot be shared at this time as the data also forms part of an ongoing study.

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
