# Peer review of "Effect of Severe Plastic Deformation and Post-Deformation Heat Treatment on the Microstructure and Superelastic Properties of Ti-50.8 at.% Ni Alloy"

_materials, 2022, doi:10.3390/ma15217822_

Round 1

Reviewer 1 Report

This paper has potential for publication in this Journal after minor revisions.

1. Why as received (AR) sample was heat-treated, then water quenched, then immersed in liquid nitrogen prior to the differential speed rolling process?

2. Please report the phase transformation temperatures and microstructure of AR sample before any treatment. 

3. Please report JCPDS cards that were used for XRD phase analysis.

4. In Fig 10, how fraction is calculated?

5. Fig. 10 should be in section 3.2.4 and similarly Fig. 11 should be in section 3.2.5.

6. Why stress for phase transformation and yield stress are not calculated from stress-strain curves rather these values are estimated from the Vicker hardness test.

Author Response

Review Report #1

  1. Why as received (AR) sample was heat-treated, then water quenched, then immersed in liquid nitrogen prior to the differential speed rolling process?

: The reason is stated in the revised paper: “The purchased plate was heat-treated at 1023 K for 15 min to relive residual stress. This sample will be referred to as the as-received (AR) sample. For applying severe plastic deformation at cryogenic temperatures, which is known to be more effective in microstructural refinement compared to SPD at room temperature [31, 33], the AR sample was immersed into liquid nitrogen for 10 minutes, removed from the liquid nitrogen bath and then immediately subjected to differential speed rolling with a speed ratio of 2:1 between the upper and lower rolls.”

  1. Please report the phase transformation temperatures and microstructure of AR sample before any treatment. 

: This is reported in Table 1.

  1. Please report JCPDS cards that were used for XRD phase analysis.

: The following is stated in caption of Figure 2 and Figure 6: Identification of phases was made based on the data from JCPDF cards (01-076-3614, 01-076-7519 and 01-076-4263).

  1. In Fig 10, how fraction is calculated?

: Figure 10 is newly provided to show how the texture component fraction was calculated.

  1. Fig. 10 should be in section 3.2.4 and similarly Fig. 11 should be in section 3.2.5.

: This is fixed.

  1. Why stress for phase transformation and yield stress are not calculated from stress-strain curves rather these values are estimated from the Vicker hardness test.

: The critical stress for martensitic transformation was calculated from the tensile tests. The following words are added: “Figure 16(a) shows the Ds calculated using the  values calculated from the  data and the critical stresses for the martensitic transformation measured from the tensile tests in Figures 13 and 15. “Sliding at the grip occurred before yielding for some cases and thus the yield stresses were calculated from the Hv data.

Reviewer 2 Report

The work has sufficient novelty and significance and can be published in a journal. The content of the work is fully consistent with the goals and objectives of the journal, I take into account the special issue. At the same time, there are a number of comments to the article that need to be addressed:

1. The introduction does not fully reflect the significant work on research in the area under consideration. For example, the recent work of Materials. 2021 Vol. 14, No. 21. P. 6256. DOI: 10.3390/ma14216256; Metals. 2022 Vol. 12, No. 7. A. 1131. DOI: 10.3390/met12071131.

2. On. fig 8 missing error bar

3. In fig 9 a, very similar diffraction patterns from martensite and Ti3Ni4 are shown, although their structures are different. It is necessary to decipher the reflexes and prove belonging to these phases.

4. Fig 9 with dark lamellas localized horizontally are Ti3Ni4 this is a fact. Horizontal light - no, this is pure monoclinic martensite. It is then necessary to remove the red arrows on the left and bring the diffraction.

5. In fig 14 c what caused the stress jump on the martensitic plateau? This cannot be a mistake, because it is also present in the reverse transition. Nothing has been written about this.

Author Response

Review Report #2

  1. The introduction does not fully reflect the significant work on research in the area under consideration. For example, the recent work of Materials. 2021 Vol. 14, No. 21. P. 6256. DOI: 10.3390/ma14216256; Metals. 2022 Vol. 12, No. 7. A. 1131. DOI: 10.3390/met12071131.

: This important reference is added in the revised version.

  1. On. fig 8 missing error bar

: Standard deviation values are added in the revised version.

  1. In fig 9 a, very similar diffraction patterns from martensite and Ti3Ni4 are shown, although their structures are different. It is necessary to decipher the reflexes and prove belonging to these phases.

: In the revised papers, this job was conducted.

  1. Fig 9 with dark lamellas localized horizontally are Ti3Ni4 this is a fact. Horizontal light - no, this is pure monoclinic martensite. It is then necessary to remove the red arrows on the left and bring the diffraction.

: This error is fixed.

  1. In fig 14 c what caused the stress jump on the martensitic plateau? This cannot be a mistake, because it is also present in the reverse transition. Nothing has been written about this.

: The following words are added: “

“It is worthwhile to note that the aged HRDSR sample (after annealing for 5 min) exhibits incremental variation in stress during repeated superelastic loading and unloading, while the other samples show a flat stress plateau. Wang et al. [42] also observed the incremental variation in stress during superelastic loading in the swaged NiTi and attributed this phenomenon to the heterogenous microstructure of the swaged sample (mixed with high- and low angle grain boundaries) where martensitic transformation occurs first at high angle grain boundaries and then later at low angle grain boundaries. The microstructure of the aged HRDSR sample (after annealing for 5 min) also consists of many high angle grain boundaries and dislocation substructure in grain interiors.”

Reviewer 3 Report

In this article, a Ni-rich NiTi alloy was processed by high-ratio differential speed rolling (HRDSR), which is a severe plastic deformation method applicable for materials in sheet form, and aging was applied to the deformed samples. The effects of grain size, dislocation density, texture and Ni4Ti3 precipitates on R-phase formation and super-elastic behavior were well examined.

1. p. 10 Fig. 4 (a) : As for the grain size, there was dominant decrease in t =5 min. Explain why it became like this.

2. p.25 References (reference list) : Write the list of references to a format of MDPI.

Author Response

Review Report #3

  1. p. 10 Fig. 4 (a) : As for the grain size, there was dominant decrease in t=5 min. Explain why it became like this.

: From Fig. 4, the grain size of the HRDSR sample (before annealing) was omitted because the grain size measured from the heavily deformed microstructure with a very high fraction of low angle grain boundaries is not meaningful.

  1. p.25 References (reference list) : Write the list of references to a format of MDPI.

:  Format was corrected.

Round 2

Reviewer 2 Report

he authors took into account my comments, I have no other questions about the manuscript.